# Photosynthesis, Phytohormone Signaling and Sugar Catabolism in the Culm Sheaths of *Phyllostachys edulis*

**DOI:** 10.3390/plants11212866

**Published:** 2022-10-27

**Authors:** Huifang Zheng, Yucong Bai, Xiangyu Li, Huajian Song, Miaomiao Cai, Zhanchao Cheng, Shaohua Mu, Juan Li, Jian Gao

**Affiliations:** Key Laboratory of National Forestry and Grassland Administration, Beijing for Bamboo & Rattan Science and Technology, International Center for Bamboo and Rattan, State Forestry and Grassland Administration, Beijing 100102, China

**Keywords:** bamboo sheath, vascular bundles, auxin, chloroplast, starch

## Abstract

Culm sheaths play an important role in supporting and protecting bamboo shoots during the growth and development period. The physiological and molecular functions of bamboo sheaths during the growth of bamboo shoots remain unclear. In this study, we investigated the morphological anatomy of culm sheaths, photosynthesis in sheath blades, storage and distribution of sugars, and the transcriptome of the sheath. Respiration in the base of the culm sheath was higher than that in the sheath blades; chloroplasts matured with the development of the sheath blades, the fluorescence efficiency *Fv/Fm* value increased from 0.3 to 0.82; and sucrose and hexose accumulated in the sheath blade and the culm sheath. The sucrose, glucose, and fructose contents of the middle sheath blades were 10.66, 5.73, and 8.84 mg/g FW, respectively. Starches accumulated in parenchymal cells close to vascular bundles. Genes related to the plant hormone signaling pathway and sugar catabolism were highly expressed in the culm sheath base. These findings provide a research basis for further understanding the possible role of bamboo sheaths in the growth and development of bamboo shoots.

## 1. Introduction

Moso bamboo (*Phyllostachys edulis*) is one of the most important non-timber forest products in the world, with the highest ecological, economic, and cultural values of all bamboo types [1]. Moso bamboo can be used as an important biomass material and a high carbon sequestration species [2,3]. The most remarkable feature of moso bamboo is its rapid growth during the bamboo shoot period. Bamboo shoots can grow to a height of 15–20 m in about 60 days [2,4]. As a fast-growing plant, moso bamboo has an annual harvest yield of 6.0–7.6 Mg C ha^−1^, indicating the high carbon sequestration potential of this species [5]. In recent years, several studies have considered the growth and development of bamboo with respect to its morphological and physiological characteristics and the molecular regulation of the bamboo culm [4,6,7]. The culm sheath is a very important organ for the growth of moso bamboo shoots. Because bamboo shoots do not have photosynthesizing leaves during high growth, and only the culm sheaths wrap the tender shoot. Regarding the energy source for bamboo shoot growth during the bamboo shoot phase, many researchers have concluded that nutrients from mature bamboo culm can be stored in the rhizome and transferred to the bamboo shoot through the rhizome [6]. However, the effect of bamboo sheaths on bamboo’s growth period is worth further study.

The thick, stiff culm sheath provides support and protection to the soft, weak culm internode during development [8]. As an important vegetative organ of bamboo shoots, the base of the culm sheath is located near the internodes (Figure 1A,B). As the bamboo sheath dries and falls off, the normal green culm inside becomes exposed (Figure 1C). Although the bamboo sheath stays on the culm for only a short time, it may be important for the normal growth of the bamboo shoot. In recent years, the effect of the culm sheath on the growth of bamboo shoots has attracted increasing attention. A study of culm sheaths in *Fargesia yunnanensis* demonstrated that, after cutting the stem sheath, the elongation of internodes slowed [9]. Previous studies in our laboratory have shown that the destruction of the culm sheath severely inhibits the normal growth of bamboo shoots and results in internode shortening [10]. Culm sheath cutting impedes the longitudinal transport of water and assimilates and restricts internode elongation. Although there have been studies on the effect of the bamboo sheath on the height growth of bamboo shoots in *Fargesia yunnanensis*, these studies mainly concerned the physiological changes in the bamboo culm after sheath stripping. *Fargesia yunnanensis* is a small bamboo, generally around 6 m high, with shoots produced from August to September, whereas *Phyllostachys edulis* is a large monopodial bamboo, up to 20 m high, with shoots produced from April to May. The size and growth rate of the two species differ considerably. Therefore, the study of the moso bamboo sheath at the anatomical, physiological, and molecular levels will be helpful to better understand its role in the growth of bamboo shoots.

Sugar is an important source of energy for plant growth and development. Studies have shown that sugar plays an important role in promoting bamboo internode elongation [11,12]. However, before height growth is completed, there are only culm sheaths on bamboo shoots and no branches or leaves [6]. During this period, bamboo shoots cannot synthesize carbohydrates through leaf photosynthesis. Some research has shown that soluble sugar and starch are used for the growth of bamboo shoots, which attached to mature bamboo by rhizomes. When bamboo forests experience low-temperature stress, mature bamboo is likely to provide immature bamboo with carbohydrates and help rebuild its light-capturing system [13,14]. In addition to relying on mature bamboo for nutrition, the role of culm sheaths in the developmental process is unknown. The stem and leaf sheaths of Graminaceae temporarily store starch, providing sufficient carbohydrates for further plant growth [15]. In addition, starch can accumulate in immature chloroplasts during leaf development in rice [16].

Although bamboo sheaths exist for a short time during the bamboo shoot stage, the sheath blade may have a leaf-like function during the development of bamboo shoots. In this study, physiological indicators, including gas exchange parameters and sugar content, were measured. The anatomical characteristics of the culm sheath were observed and analyzed, and the differentially expressed genes (DEGs) were analyzed from transcriptome data. These experiments were performed to explore the functions of photosynthesis and carbohydrate storage and metabolism in culm sheaths.

## 2. Materials and Methods

### 2.1. Study Site

The study site was located in Caicun Town, Jingxian County (N 30°44′10.464″; E 118°34′49.8504″), Anhui Province, China. Jingxian County, located in the mountainous area of southern Anhui, has a forest coverage rate of nearly 64%. Within its territory, Caicun Town is known as “the first town of Chinese bamboo” due to its bamboo garden exceeding 70,000 mu, which has been awarded the title of “Anhui Bamboo Township” by the Department of Forestry of Anhui Province. Jingxian County has a subtropical monsoonal humid climate. The mean annual precipitation is 1500 mm, and the mean annual temperature is 15 °C. The area receives an average of approximately 2113 h of sunshine and an average of 240 frost-free days per year. Shoot sheath samples were collected from March to May in 2019, 2020, and 2021.

### 2.2. Measurement of Gas Exchange Parameters and Sampling for Physiological Analysis

Gas exchange parameters were determined according to the method described by Ionuț [17]. Photosynthetic capacity was measured using a TARGAS-1 Portable Photosynthesis System (PP-Systems, Hansatech Instruments Ltd., Norfolk, UK) using culm sheaths from each experimental plant under natural conditions of 390 μmol CO_2_ mol^−1^ air and temperatures of 24–30 °C. The measurement of gas exchange parameters was divided into two parts: the first part was the detection in different parts of bamboo shoots at the base, middle, and upper culm sheaths. Measurements were made between 9:00 a.m. and 11:00 a.m. using an attached light-emitting diode (LED) as a light source. The other part was the determination of the daily variation of gas exchange parameters in the middle culm sheath of the bamboo shoot. Three culm sheaths from each shoot were selected for the measurement of the gas exchange parameters, immediately frozen in liquid nitrogen, and stored at −80 °C to determine the physiological parameters. In the shoots, 3–5, 13–15, and 23–25 pieces of the sheath (from bottom to top) were sampled, corresponding to the mature internodes, rapidly elongating internodes, and unelongated internodes of bamboo shoot internodes, respectively.

### 2.3. Chlorophyll Fluorescence Measurements

Chlorophyll fluorescence was measured using a Handy-PEA (Hansatech Instruments Ltd., Norfolk, UK) according to the method described by Brestic [18]. Between 9:00 a.m. and 11:00 a.m., we chose six bamboo shoots of approximately the same height, each with three unrolled continuous sheath blades (n = 18). Measurements of sheath blades were performed on the different sections and at different heights of bamboo shoots after leaves were dark-adapted for 15–20 min. The middle part of the sheath blade was clamped with a leaf clip for measurement.

### 2.4. Measurement of Photosynthetic Pigments

Powdered samples (0.05 g) of sheath blades from the base, middle, and top sections were placed in 15 mL centrifuge tubes, and 10 mL of 80% acetone was added. The samples were extracted in the dark at room temperature until they were completely white. *Optical density* (OD) values were measured at 470, 663, and 645 nm. The chlorophyll a and chlorophyll b contents were calculated according to the Harmut formula [19]. The experiment was repeated three times.

### 2.5. Anatomical Structure

The paraffin section method was modified from Li [4]. From a 3-m-tall bamboo sheath, 3–5 pieces of bamboo sheath located at the base of the shoot were labeled as “Base”; 13–15 pieces of the bamboo sheath located at the middle of the bamboo shoot were labeled as “Middle”; and 23–25 pieces of the bamboo sheath located at the upper part of the shoot were labeled as “Top.” Culm sheaths from different locations of the bamboo shoot were cut into approximately 1 cm × 0.5 cm samples and fixed in formalinacetic-70% alcohol (FAA, *v/v*) buffer, and gases were removed with a vacuum pump. The tissue was embedded in paraffin, after which serial transverse and longitudinal sections (8 μm thick) were cut and sequentially stained with safranin and fast green, followed by sealing with neutral gum. Sections were observed under an Olympus BX51 compound microscope. ImageJ software was used to measure the xylem area, phloem area, and thicknesses of the upper and lower epidermis. The distribution of starch grains was observed under green fluorescence using an Olympus microscope. The starch granules in freehand slices were stained with 8% iodide-potassium iodide for 1 h and then made into freehand sections for observation under a microscope. To facilitate the observation of vascular bundles at the bamboo sheath and internode, the samples were stained with 8% acid magenta for 2 h and then made into freehand sections for photographic observation under a microscope.

### 2.6. Sucrose, Glucose, and Fructose Content Determination

The contents of sucrose, glucose, and fructose were determined following the method described by Hu [20] with modifications. Samples of 0.5 g fine powder were extracted overnight with 0.7 mL of deionized water at 80 °C. The supernatants were collected by centrifugation at 12,000× *g* for 20 min. The supernatants were freeze-dried, redissolved in 1 mL deionized water, and then stored at −20 °C until analysis. Then, 10 µL extract samples were continuously incubated three times with 1 µL of glucose assay reagent (Yuanye Shanghai yuanye Bio-Technology Co., Ltd., Shanghai, China) at room temperature for 10 min, with 1 µL of phosphoglucose isomerase (Yuanye, Shanghai yuanye Bio-Technology Co., Ltd., Shanghai, China) at room temperature for 10 min, and then with 1 µL of invertase (Yuanye, Shanghai yuanye Bio-Technology Co., Ltd., Shanghai, China) at room temperature for 10 min. The absorbance was recorded at A340 nm after each incubation step to determine the glucose, fructose, and sucrose content. Each sample was measured three times.

### 2.7. Transcriptome Sequencing and Analysis

Through physiological and anatomical studies, we propose that the middle culm sheaths of bamboo shoots may be more important than the base culm sheaths and upper culm sheaths for the development of bamboo shoots. The middle culm sheaths have stronger photosynthetic and more stable vascular systems than the basal senescent culm sheaths and the upper young culm sheaths. Studies of structural and physiological indicators of different parts of individual culm sheaths also revealed large differences among the three. Therefore, the culm sheaths in the middle of the 3 m-high bamboo shoots were divided into three parts (i.e., sheath base, sheath upper, and sheath blade) for sampling. The sampling sites are shown in Figure 1D,F. All samples were immediately frozen in liquid nitrogen and stored at 80 °C. For RNA extraction and quality assessment, the samples with RIN > 8.0 were used for Illumina RNA-Seq, which was performed by Biomarker Technologies Co., Ltd. (Beijing, China) The experimental procedure was performed according to the standard protocol provided by Oxford Nanopore Technologies (ONT). The full-length sequences were identified by primer sequences at both ends of the reads and polished to obtain the consensus isoform. All consensus isoform sequences were compared to the reference genome by the minimap 2 software and then de-redundantly analyzed. The fast5 format data were converted to the fastq format after base calling by the Guppy software in the MinKNOW 2.2 package. The raw fastq data were further filtered for short fragments and low-quality reads to obtain the total clean data. The results were analyzed on the BMKCloud platform (BMKCloud www.biocloud.net (accessed between the period of March and May 2022)). The transcriptome data used in this paper have been uploaded to the National Center for Biotechnology Information under the accession number PRJNA841804.

### 2.8. qRT-PCR Verification

To verify the expression patterns of genes in different parts of the culm sheath, several genes from the starch and sugar metabolism pathways were randomly selected. qRT-PCR analysis was performed using the same transcriptome sequencing samples as the template. The tonoplast intrinsic protein 41 gene (*TIP41*) was used as the endogenous reference [21]. Primer3 (http://primer3.ut.ee/ accessed on 2 April 2022) was used to design primers, and the primer specificity check was performed using the NCBI online tool Primer-BLAST (https://www.ncbi.nlm.nih.gov/tools/primer-blast/index.cgi accessed on 2 April 2022) (Appendix A). Total RNA of the samples was extracted by TRIzol (Invitrogen, Carlsbad, CA, USA) method, and the first-strand cDNA of RNA was synthesized by using first-strand Synthesis Master Mix (LABLEAD, Beijing, China). Real-time quantitative PCR was performed using a LightCycler 480 Real Time System (Roche, Rotreuz, Switzerland). The qRT-PCR conditions were as follows: 95 °C for 10 s, 60 °C for 10 s, and 72 °C for 20 s for 45 cycles. Gene expression was evaluated three times. Gene expression was calculated using the 2^−∆∆Ct^ method [22].

### 2.9. Statistical Analysis

The data presented are the means of at least three replicates per experiment (means ± SE). One-way analysis of variance (ANOVA) and least significant difference (LSD) tests were used to determine the significance of the differences between the means. The analyses were conducted using SPSS 20.0 for Windows software. Differences were considered to be significant at *p* < 0.05. Origin software (OriginPro 2021b) was used for data mapping, and Adobe Illustrator (AI) software was used for image combination.

## 3. Results

### 3.1. Changes in the Photosynthetic Capacity of Sheath Blades

Photosynthesis is the key process necessary for plant growth. Photosynthesis in plants is generally evaluated by gas exchange parameters, including the net photosynthetic rate (Pn, μmol·m^−2^·s^−1^), transpiration rate (E, mmol·m^−2^·s^−1^), stomatal conductance (gs, mmol·m^−2^·s^−1^), and intercellular CO_2_ concentration (CO_2int_, μL·L^−1^). We measured the gas exchange parameters of 3-m-long moso bamboo shoots. The culm sheath was mainly determined by the upper part (sheath upper) and the base of the sheath (sheath base). We measured the culm sheath at the base, middle, and top of the whole bamboo shoot (Figure 1D–G). The Pn values of the sheath proper and sheath blades were negative (Figure 2A). However, the absolute value Pn of the culm sheath was significantly higher than those at the middle and top of the bamboo shoot, especially at the sheath base. This indicated that the base of the sheath displayed strong respiration, but almost no photosynthesis. The variation in water-use efficiency (WUE) was consistent with that of Pn (Figure 2E). There was no significant difference in CO_2int_ between different parts of the sample, except at the sheath base in the middle of the shoot, which was significantly higher (Figure 2B). The E of the sheath upper in the middle of the shoot was higher than that of the culm sheath base and sheath blades in the middle of the shoot (Figure 2C). The gs and E of the different samples showed similar change trends (Figure 2D). Compared with the base and the top, the chlorophyll a content of the sheath blades in the middle of the bamboo shoot was the highest (Figure 2F). Chloroplast development was more mature in the sheath blades in the middle of the shoot. The diurnal variation of the sheath blade gas exchange parameters of bamboo shoots was determined (Appendix A). The net photosynthetic rate of sheath blades measured from 17:00 p.m. to 5:00 a.m. was negative, whereas that during the day (6:30 a.m., 9:00 a.m., and 15:00 p.m.) was positive. The net photosynthetic rate was negative at 12:00 noon, suggesting that sheath blades may also have a midday depression (Appendix A). CO_2int_ continued to increase from 17:00 to 0:00 and reached its maximum at around 0:00, which may be due to the continual enhancement of respiration, leading to an increase in CO_2_ concentration in the sheath blades (Appendix A). The fluorescence value increased with an increase in bamboo shoot height (Figure 3A). As the sheath blades matured, the fluorescence parameter *Fv/Fm* gradually increased (Figure 3B). The fluorescence value of sheath blades was 0.3–0.82, indicating that, during the process of plant maturation, chloroplasts gradually matured, consistent with the chlorophyll content measured in the early stages. The Fv/Fm values of sheaths were significantly reduced under shading, indicating that shading inhibited the development of chloroplasts. Chloroplasts were observed in the parenchymal cells of the sheath blade in freehand sections (Figure 3E,F).

### 3.2. Anatomical Characteristics of Culm Sheaths

The structure of the sheath blades was different from that of the culm sheath and bamboo leaf. The thickness of the epidermal cells on the adaxial surface of the sheath blade was significantly greater than that on the abaxial surface. The abundance of vascular bundles in sheath blades and the area of xylem vessels were larger than that of the phloem, which may be more conducive to water transport (Appendix A). The cross-sectional area of the upper sheath blade phloem was smaller than that of the bottom sheath blade phloem, probably because the upper sheath blade was still developing. The vascular bundle spacing of sheath blades at the middle of the bamboo shoots was not significantly different, but it was significantly greater than that of the sheath blades at the top of the bamboo shoot. This indicated that the sheath blades at the base and middle of the bamboo shoot and the middle of the shoot were more mature than those at the top of the shoot.

The culm sheath also included epidermal, basic, and vascular tissues. Vascular bundles were arranged in a relatively ordered manner, close to the abaxial side. One side of the paraxial plane was composed of multiple layers of parenchymal cells. Along the transverse plane, the vascular bundles of the sheath alternate in size in single rows, with larger fibrous caps on the distal plane (Figure 4A–F). The vascular bundles of the sheath blades were directly connected to the vascular bundles of the sheath proper, except that the outer fibrous cap was much less developed than that of the sheath proper. The vascular bundles connecting the base of the bamboo sheath to the base of the bamboo internode were observed by acid magenta staining (Appendix A). The transport of water and assimilate between the bamboo sheath and internode depends on vascular bundles. The culm sheaths at the base of the 3-m-long bamboo shoots were shriveled and fractured, and the vascular bundles were distributed in a disorderly way, with more fibrous cells than other parts of the culm sheath of the bamboo shoot (Figure 4B). The results indicated that the culm sheath at the base of the bamboo shoot was senescent. The large and small vascular bundles were arranged at intervals with the phloem facing the outer sheath; the vascular bundles were also concentrated in the outer sheath and arranged relatively neatly, and the inner sheath was nearly full of parenchymal cells. The outer cell wall was thickened to protect the inner tissue (Figure 4C).

### 3.3. Sucrose, Glucose, and Fructose Content and Starch Distribution in Culm Sheaths

The glucose, fructose, and sucrose contents in different parts of the bamboo sheath samples were determined. For the distribution of glucose, the content in the basal bamboo sheath was lower than that in the middle and upper parts of the whole bamboo plant (Appendix A). For a single sheath, the glucose content at its base was higher than that in other parts of the sheath. The fructose content in the sheath at the base of the bamboo shoot was significantly lower than that in the other parts. The fructose content of the sheath blade in the middle of the bamboo shoot was higher than that of the sheath blades at the base and the top of the bamboo shoot. The fructose content in the culm sheath at the top of the bamboo shoot was significantly higher than that at the base and middle of the bamboo shoot (Appendix A). The sucrose content was the highest in the middle sheath, followed by the upper sheath, and the remaining parts did not differ significantly with respect to sucrose content (Appendix A). Starch granules were observed in the culm sheaths and sheath blades under fluorescence microscopy. Starch grains were not observed in the bamboo sheath at the base of the bamboo shoot (Figure 5A–C). Starch grains were observed in the sheath in the middle and top of the bamboo shoot, but more starch grains were observed in the upper sheath, especially at the sheath base (Figure 5G–I). The basal sheath corresponded to the mature nodes of the shoots, and starch grains were less distributed. There were more starch granules in the middle and upper shoots. Starch grains were mainly distributed in parenchymal cells around the sheath of vascular bundles, and there were few or no starch granules in parenchymal cells far from the vascular bundles (Figure 5J,K). The potassium iodide staining results in the freehand sections were consistent with those observed under fluorescence (Figure 5L–O).

### 3.4. Active Phytohormone Signaling and Sugar Catabolism in the Sheath Base

The transcriptome of different parts of the culm sheath in the middle of bamboo shoots (sheath base, sheath upper, and sheath blade) was sequenced. Clean data from sequencing each sample reached 2.12 GB. The full-length rate reached more than 89%, and the clean reads obtained from transcriptome sequencing were matched with the reference genome transcripts between 94.59% and 99.10%, which indicates that the quality of the RNA-seq data is reliable. A total of 12,964 differentially expressed genes (DEGs) were identified, with at least a 1.5-fold change and adjusted *p*-value ≤ 0.05 using DESeq2. KEGG class and enrichment analysis of DEGs showed that phytohormone signaling, starch and sucrose metabolism, and photosynthesis-related pathways may be more important in different parts of the culm sheath (Appendix A).

Plants coordinate responses to complex environmental stresses through a variety of hormonal signaling pathways, including auxin, cytokinin, gibberellin, abscisic acid, ethylene, brassinosteroid, jasmonic acid, and salicylic acid (Figure 6). There were 57 genes, including 2, 34, 4, 4, and 13 linked to auxin influx carrier (AUX1), AUX indoleacetic acid (IAA)-induced protein (AUX/IAA), auxin response factor (ARF), auxin responsive Glycoside hydrolase 3 (GH3) gene family, and small auxin upregulated RNA (SAUR) family proteins, respectively, involved in tryptophan metabolism. Most of these genes had higher expression levels in the sheath base than in the sheath upper and sheath blades. The identified AUX1, ARF, and GH3 were highly expressed in the culm sheath base, except for *PH02Gene13076* of GH3. Among the 34 AUX/IAA genes, 24 genes had a higher expression level in the sheath base than in the sheath upper and sheath blade, and only five genes (*PH02Gene41082*, *PH02Gene36936*, *PH02Gene40027*, *PH02Gene45304*, *PH02Gene50429*) had a higher expression level in the sheath blade than in other parts. Four of the eighteen genes were highly expressed in the sheath base, and most were highly expressed in the sheath upper. Two (*PH02Gene17323*, *PH02Gene13923*) of the four genes were highly expressed in the sheath blade, and the rest (*PH02Gene04304*, *PH02Gene36869*) were highly expressed in the base of the sheath in diterpenoid biosynthesis. In the other five hormone signal transduction pathways, most genes were highly expressed in the base or segment of the sheath, and a few genes were highly expressed in the upper part of the sheath. The expression levels of genes involved in these pathways showed that the expression levels of most of the genes related to hormone signal transduction were relatively high in the sheath base, as well as in the sheath blade. In particular, the hormones associated with cell elongation include auxin, brassinosteroids, and jasmonic acid.

A total of 97 genes were identified in the starch and sucrose metabolic pathways (Figure 7). Among them, starch synthase (STS), hexokinase (HXK), fructokinase (FRK), and invertase (INV) gene expression in the sheath base were higher than that in the sheath upper and sheath blade. Three sucrose phosphatase (SPS) genes were expressed in the sheath blade, and these showed a high expression level in the sheath base and sheath upper. Another 14 genes showed high expression levels in sheath leaves. The sheath base and sheath blade of the bamboo sheath had a high energy demand and vigorous metabolic activity. A total of 57 genes were mapped to each stage of photosynthesis, most of which were highly expressed in culm sheaths, and some genes of photosystem I had relatively higher expression levels in the upper culm sheaths compared with the culm sheath base and sheath blade (Appendix A).

To further confirm these findings, several candidate genes of plant hormone signal transduction pathways and starch and sugar metabolism pathways were randomly selected for qRT-PCR analysis, indicating that the hormone signaling pathway regulates very important growth and development processes at the base of bamboo shoots.

## 4. Discussion

Moso bamboo grows rapidly and can accumulate a large amount of biomass in a short time. Culm sheaths play a very important role in the growth of bamboo shoots. In this study, we revealed the function of the sheath blade and culm sheath from physiological, anatomical, and molecular perspectives.

### 4.1. Sheath Blades Are Capable of Photosynthesis and This Ability Develops Gradually

Fluorescence parameters are often combined with photosynthetic gas exchange to study the regulatory processes of leaf photosynthesis [23]. The increase in chlorophyll content and *Fv/Fm* value indicated that chloroplast function gradually improved with the development of sheath blades. In our study, sheath blades in the middle of bamboo shoots had a relatively strong photosynthetic capacity. Diurnal changes in gas exchange parameters are related to changes in environmental conditions, such as temperature, light, and CO_2_ concentration [24,25,26,27]. Our results indicate that sheath blades have a photosynthetic capacity during daytime and have midday depression. Many chloroplasts were distributed in the parenchymal cells of sheath blades, which provided a structural basis for further elucidating the photosynthetic capacity of sheath blades. Transcriptome data showed that most of the genes involved in photosynthesis, including photosystem I and photosystem II, were highly expressed in sheath blades. A few genes were highly expressed in the upper part of the sheath. The expression level of photosynthesis-related genes was low in the base of the culm sheath. This is probably because the base of the culm sheath is surrounded by adjacent culm sheaths and cannot receive light. Light conditions determine the photosynthetic physiological characteristics of leaf sheaths [28].

In this study, the sucrose content in sheath blades was higher than that in the upper and base of the culm sheaths. Sucrose produced by photosynthesis can be transported over long distances in vascular bundles. Sucrose phosphate synthase is a rate-limiting enzyme in sucrose synthesis that is irreversibly catalyzed by UDP-glucose and fructose-6-phosphate [29]. In sheath blades, the three SPS genes were all highly expressed, indicating that in sheath blades, synthesized photosynthates were transformed into sucrose and transported to the sink tissue. In the node, the large number of bifurcated vascular bundles form a complex network system, which can not only transport the photosynthate to the next segment but also transport the photosynthate from one side to the other [30]. Whether the sucrose synthesized by culm sheaths can be transported to bamboo shoots through the node for metabolic use by the shoot needs to be verified by direct experiments with ^13^C.

### 4.2. Sheath Base Stores Starch and Is the Main Site of Respiration and Sugar Metabolism

Starch grain storage and sugar metabolism are crucial for the growth and development of shoot sheaths and bamboo shoots. Young culm sheaths store more starch grains than mature culm sheaths. As the sheaths matured, the starch in the parenchymal cells distant from the vascular bundle was first consumed. The glucose produced by starch metabolism can be utilized by organisms as an energy substance and as an effective signaling molecule, and it plays an important role in all stages of the plant life cycle, from germination and vegetative growth to reproductive development and seed formation [29,31]. Glucose content is high at the base of the sheath, where the tissues are tender, and the parts with active cell metabolism often need more sugar and can be used directly. The more active the cell division, the higher the sugar content [11]. However, glucose metabolism did not play a major role in internode elongation, and water pressure, sucrose, and starch hydrolysis were the main causes of internode elongation [12]. Higher water inflow, sugar transport, and catabolism increase the swelling pressure of the cells, which then promotes cell expansion and internode elongation.

To maintain rapid growth, bamboo shoots need a large amount of carbohydrate metabolism to support cell division and elongation activities. Soluble sugars, such as sucrose, glucose, and fructose, not only regulate plant metabolic activities as substrates of energy metabolism but also regulate the expression of many key genes as signal molecules and thus participate in the regulation of plant growth and development. In our analysis of starch and sucrose metabolic pathways determining the expression changes in DEGs, the sucrose decomposition genes (INV, sucrose synthase (SUS), HXK, FRK) in the sheath base and the sucrose synthesis genes (SPS) in the sheath blade were highly expressed. This is consistent with the low sucrose content and high glucose content in the sheath base and the high sucrose content and low glucose content in the sheath blade. Glucose is not only a direct substrate for glycolysis but is also a signal molecule that regulates plant growth and development [32,33]. *Arabidopsis thaliana* hexokinase 1 (HXK1) kinase is a glucose receptor that integrates nutritional and hormonal signals to regulate gene expression and plant growth and development [34]. In *Arabidopsis thaliana*, photosynthesis is mainly controlled by glucose through glycolysis and mitochondrial bioenergy transmission stimulation to control the TOR signal to quickly control the metabolic transcription network and activate the cell cycle in the root meristem [32]. How glucose plays a signaling role in regulating the growth and development of bamboo shoots requires further study.

### 4.3. Plant Hormones Play an Important Role in Culm Sheaths

The development of the culm sheath may affect the rapid elongation of bamboo internodes [12,35]. The rapid growth of bamboo is mainly due to the vigorous division and rapid elongation of internode cells, and plant hormones play a very important role in plant growth and development. Our group has reported for the first time that the culm sheath is the main site of auxin synthesis, transporting synthesized auxin to the connected internodes and thus regulating internode elongation [36]. Endogenous brassinosteroids (BRs) are the main factors promoting internode elongation in bamboo shoots [37]. Both auxin and BRs are involved in plant growth and development by the dissolution and reconstruction of cell walls, regulating cell division [38]. The moso bamboo gibberellin-stimulated transcript (GAST) protein is thought to have a unique regulatory role in the ABA pathway for rapid shoot growth and abiotic stress responses [39]. Culm sheaths may affect the growth and development of bamboo shoots through multiple hormone-signaling transduction pathways.

The base of the culm sheath has stronger respiration and a higher glucose content compared to in the middle of the culm sheath and the top of the culm sheath, while the base of the culm sheath is connected with the base of the internode near the site of the intermediate meristem [40]. The development, maturation, and aging of bamboo culms are initiated from the base, followed by the middle and top bamboo culms [41]. The absolute value of the net photosynthesis of the culm sheath near the base of the bamboo shoot was lower than that of the middle and upper culm sheaths, indicating that the respiration intensity of the culm sheath was weak in this part due to its senescent state. Senescence of shoot sheaths is a multi-level regulatory process that includes sugar accumulation mediated by a hexokinase gene, such as the *HXK6* trigger, and temporary expression of transcription factors *NAC2* and *WRKY75*, which promote the exacerbation of senescence, leading to chloroplast decay and reactive oxygen species, abscisic acid, and salicylic acid accumulation [35]. Senescence of culm sheaths stimulates the bamboo shoot to redistribute carbohydrates in the tissue, transporting sugar to the bamboo shoot for reuse.

## 5. Conclusions

Culm sheaths were capable of photosynthesis, and their photosynthetic capacity gradually increased as the sheath developed due to an increase in chlorophyll content in the sheath blade. The abundance of vascular structures in the culm sheath tissue facilitated the transport of water and material. Compared to the upper part of the culm sheath and the sheath blade, the base of the culm sheath was mainly responsible for starch storage and was the main site of respiration and metabolism. It was also the main site of hormone signaling and sugar catabolism, and the transcriptional data supported these findings. The results of this study provide detailed data for a better understanding of the structure and function of culm sheaths.

## Figures and Tables

**Figure 1 plants-11-02866-f001:**
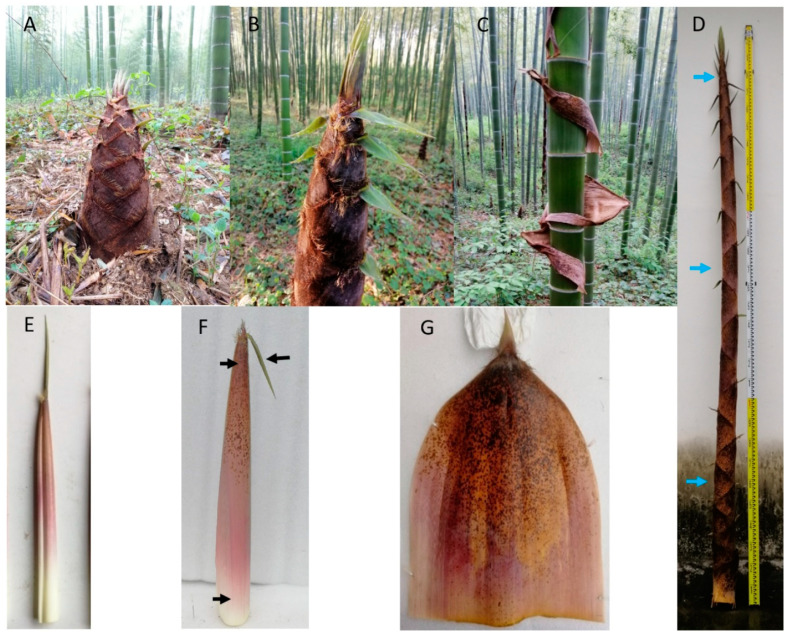
Culm sheaths of moso bamboo shoots at different stages of development and the morphological characteristics of the bamboo sheath. (**A**) Culm sheaths of early bamboo shoots. (**B**) Culm sheaths of a rapidly growing bamboo shoot. (**C**) Culm sheaths in the process of shedding. (**D**) Culm sheaths of 3-m-long bamboo shoots. (**E**) Culm sheaths in the upper parts of 3-m-long bamboo shoots. (**F**) Culm sheaths in the middle of 3-m-long bamboo shoots. (**G**) Culm sheaths in the base of 3-m-long bamboo shoots. Blue arrows indicate sampling locations on bamboo shoots, and black arrows indicate sampling locations on culm sheaths. For bamboo shoots, the sampling sites are “Top”, “Middle”, and “Base” (in order from top to bottom). For a single bamboo sheath, the sampling sites are “sheath base”, “sheath upper”, and “sheath blade” (from bottom to top).

**Figure 2 plants-11-02866-f002:**
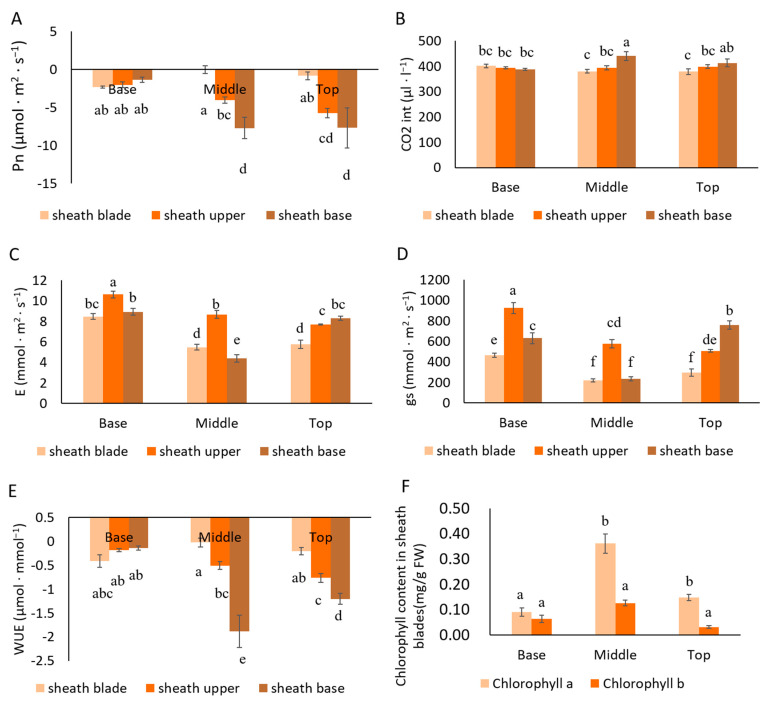
Gas exchange parameters of bamboo sheaths and chlorophyll content in sheath blades. (**A**) Pn: net photosynthetic rate. (**B**) CO_2int_: intercellular CO_2_ concentration. (**C**) E: transpiration rates. (**D**) gs: stomatal conductance. (**E**) WUE: water-use efficiency. (**F**) Chlorophyll content in different parts. The experiments were repeated three times. Different letters indicate significant differences (*p* < 0.05).

**Figure 3 plants-11-02866-f003:**
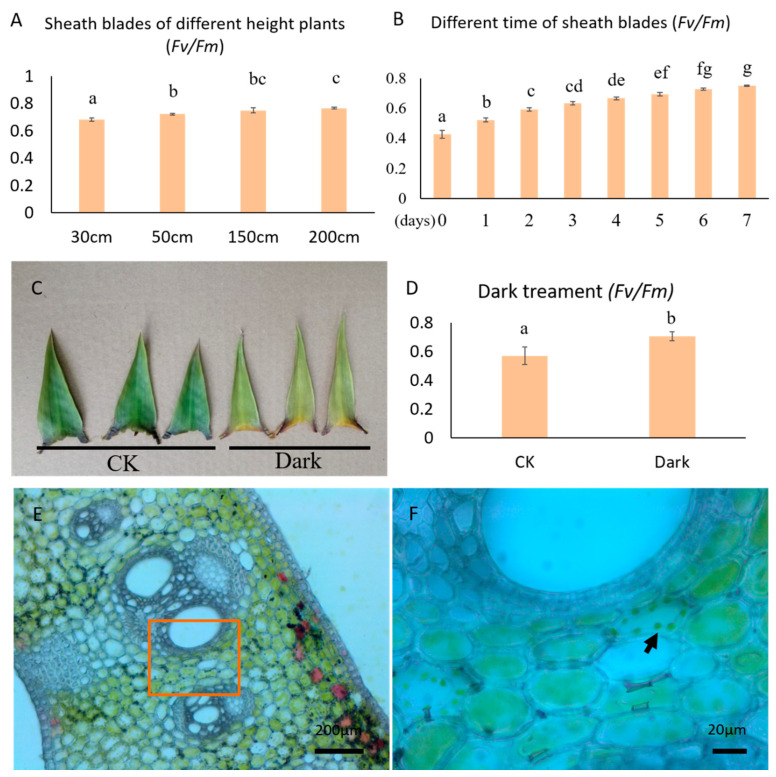
Changes in fluorescence parameters. (**A**) *Fv/Fm* values for sheath blades located in the middle of bamboo shoots at different heights. (**B**) *Fv/Fm* value of newly expanded sheath blades measured for 8 consecutive days. (**C**,**D**) *Fv/Fm* value of shaded bamboo sheaths. ‘CK’ indicates that the culm sheaths were not shaded, and ‘Dark’ indicates that the culm sheaths were shaded with black cloth. (**E**) Chloroplast distribution in sheath blades. (**F**) Enlargement of the red rectangle in (**E**). The black arrows indicate chloroplasts. Different letters indicate significant differences between the average values of the column (±SE, n = 6), *p* < 0.05.

**Figure 4 plants-11-02866-f004:**
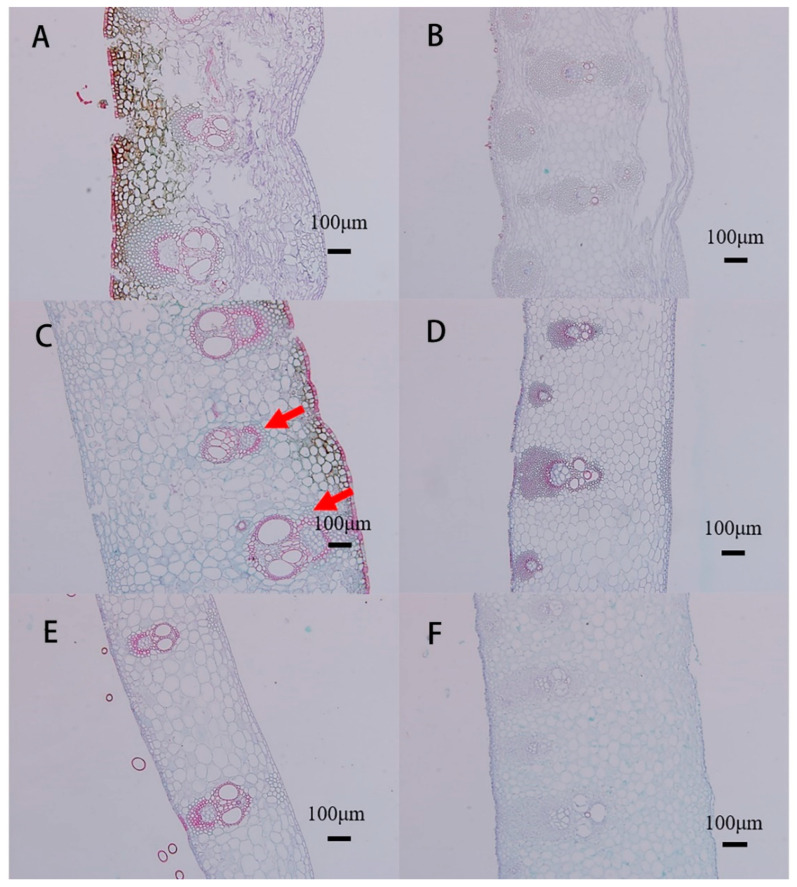
Anatomy of the culm sheaths. (**A**,**B**) Sheath upper and sheath base of the culm sheath at the base of the bamboo shoot. (**C**,**D**) Sheath upper and sheath base of the culm sheath at the middle of the bamboo shoot. (**E**,**F**) Sheath upper and sheath base of culm sheath at the top of the bamboo shoot, large and small vascular bundles (red arrow).

**Figure 5 plants-11-02866-f005:**
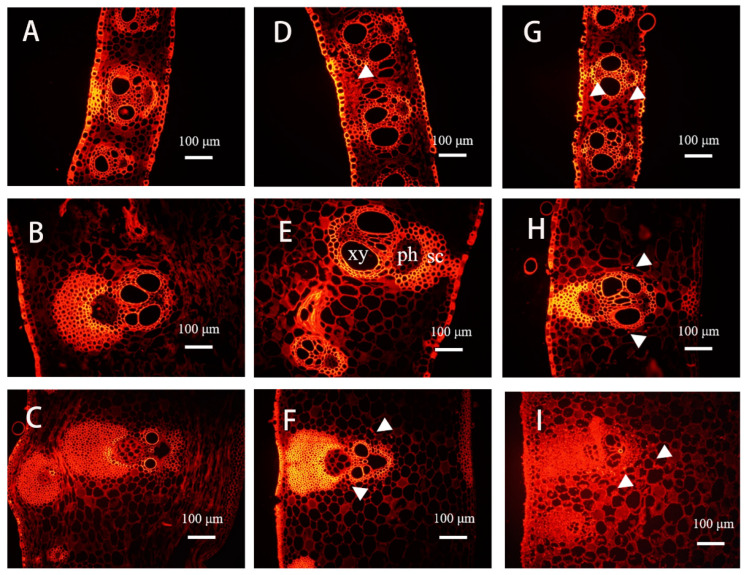
Distribution of starch grains in the culm sheath under a fluorescence microscope; potassium iodide stain was used to indicate starch. (**A**–**C**) Sheath blades, sheath upper, and sheath base of culm sheath at the base of bamboo shoot. (**D**–**F**) Sheath blades, sheath upper, and sheath base of culm sheath at the middle of bamboo shoot. (**G**–**I**) Sheath blades, sheath upper, and sheath base of culm sheath at the upper of bamboo shoot. (**J**,**K**) Longitudinal section of a culm sheath of parenchymal cells surrounding the vascular bundles. (**K**) Parenchymal cells located away from vascular bundles. (**L**) Sheath blades. (**M**) Sheath upper. (**N**) Sheath base. (**O**) Local amplification of N. xy, xylem; ph, phloem; sc, sclerenchyma. The white triangles indicate starch grains. The red rectangle indicates vascular bundle. The red arrows indicate starch grains.

**Figure 6 plants-11-02866-f006:**
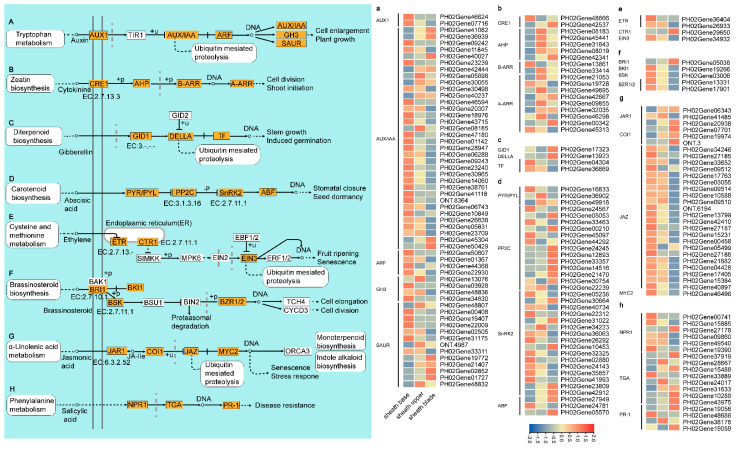
Expression changes in plant hormone signal transduction pathway-related DEGs in the culm sheath. (**A**–**H**) Schematic diagram of plant hormone signal transduction. (**a**–**h**) Heatmap of genes related to plant hormone signal transduction. The DEGs in this study are indicated by colored, shaded rectangles.

**Figure 7 plants-11-02866-f007:**
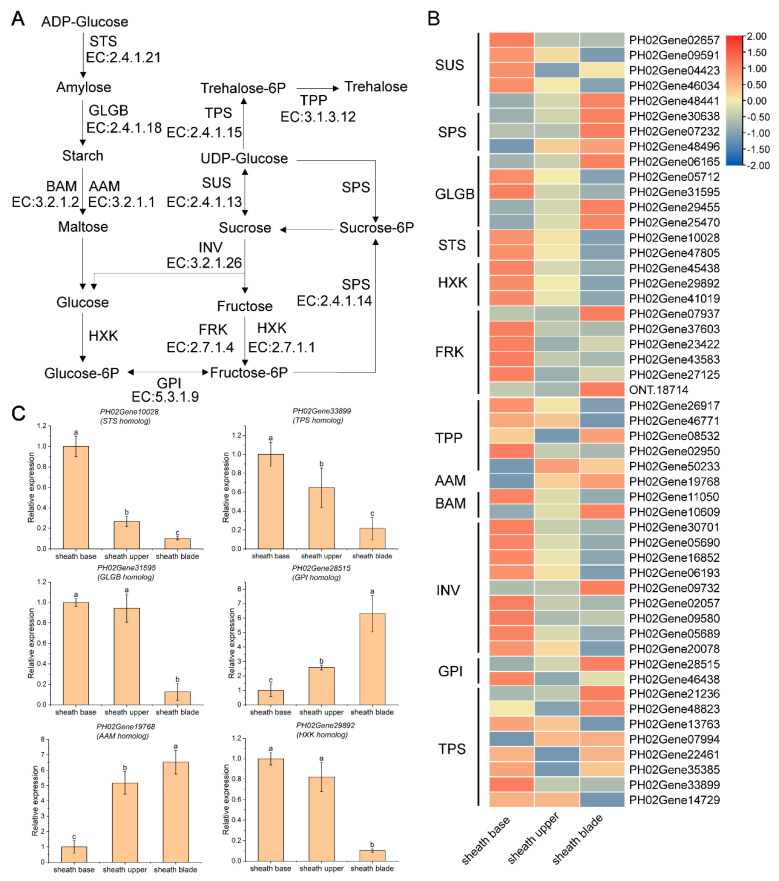
Expression changes in starch and sucrose metabolism pathway-related differentially expressed genes (DEGs) in the culm sheath. (**A**) Schematic diagram of starch and sucrose metabolism. (**B**) Heatmap of differential expression of key enzymes in starch and sucrose metabolism. STS, starch synthase; GLGB, 1,4-alpha-glucan branching enzyme; AAM, alpha-amylase; BAM, beta-amylase; HXK, hexokinase; GPI, glucose-6-phosphate isomerase; FRK, fructokinase; INV, beta-fructofuranosidase; SUS, sucrose synthase; SPS, sucrose-phosphate synthase; TPS, trehalose 6-phosphate synthase/phosphatase; TPP, trehalose 6-phosphate phosphatase. (**C**) RT-qPCR analysis of the expression pattern of specific DEGs of starch and sucrose metabolism pathway. The ID name of each DEG is placed at the top of the panel. Different letters indicate significant differences between the average values of the column (±SE, n = 3), *p* < 0.05.

## Data Availability

Publicly available datasets were analyzed in this study. This data can be found here: The National Center for Biotechnology Information under the accession number PRJNA841804.

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
