# Peer review of "Photosynthesis, Phytohormone Signaling and Sugar Catabolism in the Culm Sheaths of Phyllostachys edulis"

_plants, 2022, doi:10.3390/plants11212866_

Round 1

Reviewer 1 Report

The research content of the manuscript is very interesting, but some research content needs to be improved. I have some suggestions for the author to consider.

*The writing of the manuscript needs to be further strengthened. Some sentences are not professional enough to understand. The font of all contents shall be consistent.

*Heatmaps of gene expression levels should be subjected to hierarchical clustering analysis.

*In the pathway diagram, the corresponding number of each enzyme shall be marked.

*The phytohormone content of sheath base, sheath upper, and sheath blade needs to be measured.

Reviewer 2 Report

The paper is very nicely written and can be published as such how ever the abstract needs modifications as  numerical values of results are missing in abstract.

Reviewer 3 Report

The manuscript entitles “Photosynthesis, phytohormone signaling and sugar catabolism in the culm sheaths of Phyllostachys edulis” has been written good and have some comments below:

In abstract, Line no 12: The sentence should be remove, “. However, few studies have been conducted on the anatomical, physiological and molecular levels of the moso bamboo” and give what is important for your study.

In abstract, kindly provide some data you observed in your study.

Keywords should be different from the title.

Line no 31: In Introduction part please clear what do you want to say for “between 6.0 Mg/ha and 7.6 Mg/ha o”

Line no 38: The effect of bamboo 38 sheaths on bamboo’s special growth period is worth further study. Please font size should be same thorough out the paper.

Line number 45 to 75, Font size should be same.

In fig 1, line no 250 to 252, was repeated  as it is also written inside the text. So kindly rectify and rewrite.

In fig 2, what do you mean the letters on error bar? Please write about these letters in figure legend.

In fig 3, please write about CK as you have written in the figure.

Fig 6, should be clear.

In Reference no 6, line no 541 Phyllostachys heterocycla should be in italics.

References should be according to the journals guideline. And follow same format for all the references.

English language must be improved in throughout the manuscript.

Round 2

Reviewer 1 Report

No more comments!